# NF1 Patients Receiving Breast Cancer Screening: Insights from The Ontario High Risk Breast Screening Program

**DOI:** 10.3390/cancers11050707

**Published:** 2019-05-22

**Authors:** Nika Maani, Shelley Westergard, Joanna Yang, Anabel M. Scaranelo, Stephanie Telesca, Emily Thain, Nathan F. Schachter, Jeanna M. McCuaig, Raymond H. Kim

**Affiliations:** 1Program for Genetics and Genome Biology, Hospital for Sick Children, Toronto, ON M5G 1X8, Canada; nika.maani@sickkids.ca; 2Familial Breast and Ovarian Cancer Clinic, Princess Margaret Hospital Cancer Centre, Toronto, ON M5G 2C1, Canada; emily.thain@uhn.ca (E.T.); jeanna.mccuaig@uhn.ca (J.M.M.); 3Department of Molecular Genetics, University of Toronto, Toronto, ON M5S 1A8, Canada; joannay.yang@mail.utoronto.ca; 4Average and High-Risk Ontario Breast Screening Program, Princess Margaret Cancer Centre, Toronto, ON M5G 2C1, Canada; shelley.westergard@uhn.ca; 5Department of Medical Imaging, University of Toronto, Toronto, ON M5T 1W7, Canada; anabel.scaranelo@uhn.ca; 6Joint Department of Medical Imaging, Division of Breast Imaging, Princess Margaret Cancer Centre, Sinai Health System and Women’s College Hospital, Toronto, ON M5G 2C1 and M5S 1B2, Canada; 7Prenatal Diagnosis and Medical Genetics Program, Mount Sinai Hospital, Toronto, ON M5G 1X5, Canada; stephanie.telesca@sinaihealthsystem.ca; 8University Health Network, Toronto, ON M5GC4, Canada; nathan.schachter@uhn.ca; 9Familial Breast and Ovarian Cancer Clinic, Division of Medical Oncology and Hematology, Princess Margaret Cancer Centre, Toronto, ON M5G 2C1, Canada; 10Department of Medicine, University of Toronto, Toronto, ON M5S 1A8, Canada

**Keywords:** breast cancer, neurofibromatosis type I, high-risk screening, screening uptake

## Abstract

Neurofibromatosis Type I (NF1) is caused by variants in neurofibromin (*NF1*). NF1 predisposes to a variety of benign and malignant tumor types, including breast cancer. Women with NF1 <50 years of age possess an up to five-fold increased risk of developing breast cancer compared with the general population. Impaired emotional functioning is reported as a comorbidity that may influence the participation of NF1 patients in regular clinical surveillance despite their increased risk of breast and other cancers. Despite emphasis on breast cancer surveillance in women with NF1, the uptake and feasibility of high-risk screening programs in this population remains unclear. A retrospective chart review between 2014–2018 of female NF1 patients seen at the Elizabeth Raab Neurofibromatosis Clinic (ERNC) in Ontario was conducted to examine the uptake of high-risk breast cancer screening, radiologic findings, and breast cancer characteristics. 61 women with pathogenic variants in *NF1* enrolled in the high-risk Ontario breast screening program (HR-OBSP); 95% completed at least one high-risk breast screening modality, and four were diagnosed with invasive breast cancer. Our findings support the integration of a formal breast screening programs in clinical management of NF1 patients.

## 1. Introduction

Neurofibromatosis type 1 (NF1) is an autosomal dominant, multi-system genetic disorder affecting 1 in 3000 individuals worldwide. Causative germline variants in the NF1 tumor suppressor gene are 100% penetrant; however, the wide variable expressivity, rate of progression, and disease severity that exists among NF1 patients makes disease management challenging [1,2]. The disorder is usually diagnosed by physical examination and assessment of family history, although molecular genetic testing options are available and widely used [3]. The principal features of NF1 are cutaneous features such as café-au-lait macules, axillary/inguinal freckling and neurofibromas, which are benign nerve-sheath tumors of the peripheral nervous system [4]. Other manifestations include Lisch nodules, low-grade gliomas, skeletal abnormalities [5] and involvement of various other organ systems [6]. Notably, NF1 predisposes affected individuals to several types of malignant neoplasms [7,8] and recently several studies have established an association between NF1 and an increased risk for breast cancer [9,10,11,12,13,14].

NF1-associated breast cancers typically arise at a younger age and have the highest incidence in women under 40 years of age; women with NF1 less than 50 years of age have been recently identified to have an up to five-fold increased risk of breast cancer [14]. After age 50, breast cancer risk in women with NF1 does not differ significantly compared to women in the general population [7,8,15]. Importantly, NF1 is associated with relatively poor breast cancer survival [7,14]. Due to these findings, enhanced breast cancer screening is recommended for NF1 patients [15]. Published guidelines from the National Comprehensive Cancer Network (NCCN) recommend annual breast screening with mammography. With consideration of tomosynthesis beginning at age 30 and consideration of breast magnetic resonance imaging (MRI) from ages 30–50 [16]. Currently evidence is insufficient to recommend risk reducing mastectomy based solely on a diagnosis of NF1.

Among NF1 patients, breast cancer may be difficult to detect by examination or conventional mammography due to the presence of cutaneous neurofibromas, which have been reported to appear as oval/round nodules with well-defined borders on mammogram and may therefore hide the presence of other benign or malignant neoplastic lesions. This can lead to false-positive screening mammography recalls [17] interfering with the interpretation digital mammography without tomosynthesis results, and impede the palpation of significant tumors during clinical breast exams [18,19]. To subvert the limitations of traditional breast cancer screening with mammography, supplementary screening with breast MRI, or ultrasound when breast MRI cannot be done, have been recommended in this patient population [19]. NF1 patients are known to experience higher levels of depression and anxiety, which may diminish their long-term participation in clinical surveillance and screening programs [20,21,22]. Consequently, many cases of NF1-associated breast cancer are not diagnosed until advanced stages [13]. Thus, worse breast cancer prognosis in NF1 patients may not be a characteristic of the disease itself but may instead result from late-stage diagnosis.

The increased risk of breast cancer in NF1 has now been reported by numerous groups, including a recent review of the literature which examined 32 published epidemiological studies focused on the potential association of NF1 and increased breast cancer risk [9,10,11,12]. Interestingly, Frayling et al., provided evidence that breast cancer risk in NF1 patients may be limited to certain NF1 variants [23], suggesting that only a subset of women may require enhanced screening. Nevertheless, high-risk breast cancer screening is currently recommended for all women with NF1; however, there is a lack of data regarding the feasibility and effectiveness of this screening. Several studies have demonstrated that dysregulated emotional functioning in NF1 patients may influence the willingness of these women to participate in continued clinical surveillance, despite their increased risk of breast and other cancers [20,21]. Impaired emotional functioning (i.e., self-reported depression, anxiety, stress and self-esteem) has also been as a significant comorbidity in NF1 [20,22,24]. To address this gap in the literature, we sought to evaluate the level of uptake with recommended breast cancer screening among women with NF1 enrolled in a government-funded cancer screening program.

In the province of Ontario, breast cancer screening with annual mammography and breast MRI is facilitated through an organized provincial high-risk screening program called the High-Risk Ontario Breast Screening Program (HR-OBSP). Through the HR-OBSP, high-risk women are assessed through the use of digital mammography and breast MRI (or bilateral screening breast ultrasound, if MRI is not feasible) on the same day or within a week of time between tests, providing early detection of breast cancer in high-risk Ontario women [25]. Knowledge of the characteristics of radiological findings which could lead to the detection of early, late, and/or multiple breast cancers in the NF1 population is already established from other high-risk populations undergoing high-risk screening at our institution [26]. All women aged 30–69 with a confirmed *NF1* gene variant followed at the Elisabeth Raab Neurofibromatosis Clinic (ERNC) in the University Health Network (UHN) are offered an HR-OBSP referral. Enrolment into the HR-OBSP is limited to confirmed carriers of pathogenic variants. Consequently, patients with solely a clinical diagnosis and/or no pathogenic *NF1* gene variant are not screened through the HR-OBSP. Herein, we report on the uptake and effectiveness of breast screening through the HR-OBSP among women with NF1 to determine the feasibility of routinized high-risk screening in this population.

## 2. Results

### 2.1. Patient Population

461 unrelated NF1 cases were seen at the ERNC between 2014 and 2018. Among these cases, 245 (53%) were female, 105 (43%) of whom qualified for referral to the HR-OBSP. Of the 140 (57%) women not referred to HR-OBSP, 75 were <30 years of age. As enrolment into the HR-OBSP is limited to patients with a confirmed pathogenic variant in *NF1*, six women were excluded from the study because they declined genetic testing and 59 women were excluded because they tested negative for pathogenic variants in *NF1*. Methods for *NF1* germline genetic analysis included Sanger sequencing, multiplex ligation probe amplification, next generation sequencing, and/or RNA sequencing for deep intronic variants.

Genetic testing was administered based on the availability of genetic testing technology at the time of diagnosis. Of the 105 women who qualified for HR-OBSP referral, 28 were transferred to an outside same program facility, seven were unable to be reached by the nurse navigator, two declined HR-OBSP enrolment at the time of referral, and one woman was diagnosed with breast cancer prior to HR-OBSP referral. Of the 67 (64%) remaining women, 6/67 (9%) had screens pending at the time of referral, leaving 61/67 (91%) enrolled in HR-OBSP at UHN (Figure 1).

### 2.2. Imaging Findings

A total of 61 women with NF1 ranging from 30 to 67 years of age (median 40) were enrolled in high-risk breast screening through the HR-OBSP from January 2014 to December 2018. A total of 112 screening full-field digital mammography, three supplementary screening with breast ultrasound and 108 supplementary screening with breast magnetic resonance imaging tests were performed during the screening period. Dense breast tissue (American College of Radiology mammographic density C and D) was reported in 71% of patients during baseline screen and moderate or marked background parenchymal enhancement in 59% of patients at the time of the initial breast MRI.

A total of 95 imaging findings classified as “typically benign” or “incomplete screening for diagnostic assessment”, were reported during the screening period amongst our cohort of 61 NF1 patients enrolled in the HR-OBSP. 37/95 (39%) of these imaging findings were considered to be actionable. Actionable is defined as an imaging finding that requires further imaging investigations and/or percutaneous breast biopsies. Of the total 95 imaging findings, the majority 70/95 (74%) were described by MRI, 2/95 (2%) by ultrasound and 7/95 (7%) by mammography. The most common imaging finding reported was well-circumscribed cutaneous masses in all imaging modalities. The presence of masses in the fibroglandular breast tissue, with either a solid or cystic nature, were the second more frequent imaging finding reported.

Of the 37 actionable imaging findings, 15/37 (41%) findings were further evaluated by percutaneous imaging-guided breast biopsy. There were 10 actionable imaging findings recalled by mammography: one distortion, one mass lesion, four calcifications, three asymmetries and one other. Breast MRI showed 27 actionable imaging findings: eight mass lesions, eight non-mass enhancements, two focus and nine other. The mammography recall rate was 9% (10/112) and the MRI recall rate was 25% (27/108) (Table 1).

Table 1 lists the prevalence of the actionable findings distributed by imaging modality. A total of 12 women underwent a percutaneous imaging-guided breast biopsy of 15 actionable imaging findings (three patients had more than one needle biopsy) and seven women underwent surgical treatment. 63% of biopsies were guided by ultrasound and the remaining 38% were equally distributed among either MRI or stereotactic (mammographic) guidance. In total, four women received a pathologic diagnosis of a breast cancer on both needle biopsy and surgical pathology. The other three patients undergoing surgery had high-risk lesions and one benign papillary lesion. All four cancers were detected by breast MRI in the initial screening round (prevalence screening, 61 examinations). One cancer was also concomitantly depicted by mammography, presenting as mammographic distortion. Pathology results showed one microinvasive and three high/intermediate nuclear grade invasive ductal carcinomas; all were HER2 negative cancers. Benign findings in the remaining percutaneous biopsies were fibrocystic changes, normal parenchymal lesions (i.e., fibromas of the overlying skin), ductal epithelial hyperplasia, benign sebaceous/epidermal cysts, benign papillary lesions, benign intramammary lymph nodes, fibroadenomas, fibrocystic adenosis, or benign epithelial proliferative changes. Overall, the preliminary observation of the cancer detection incidence in this cohort could be considered as high as 65.76 per 1000 cases (prevalent screening) or as low as 35.39 per 1000 cases (overall screening to date).

### 2.3. Patient Screening Uptake

Patient screening uptake was evaluated for all female NF1 patients who were referred to the HR-OBSP at UHN and seen by the nurse navigator. Women who completed at least one breast screening test (i.e., imaging modality) through the HR-OBSP during the study period were considered primary screen compliant; women who completed more than one breast screening test (i.e., imaging modality) through the HR-OBSP during the study period were considered secondary screen compliant (Table 2). At the time of this study, 6/67 (9%) women referred to the HR-OBSP and seen by the nurse navigator had their primary screen on hold for clinical indications (breastfeeding, pregnancy and regulation of menstrual cycles). High-risk screening uptake was high among the remaining 61 women (Figure 1, Table 2); however, one (2%) woman was non-compliant at the start of the scheduling process and did not schedule any high-risk screening. Of the remaining 60 women scheduled for high-risk screening, 3/60 (5%) were non-compliant after scheduling, and did not complete any of their scheduled primary screens. Among 57 women who completed a primary screen, 55/57 (97%) completed secondary screens. 27/55 (49%) women were recalled to evaluate abnormal changes identified on primary and/or secondary high-risk breast screens, warranting a diagnostic evaluation to assess for early detection of breast cancer or atypical changes in the breast; all 27 women completed their recall assessments. Ultimately, 4 (7%) women out of the 57 women who completed at least one screen through the HR-OBSP were diagnosed with invasive breast cancer (Figure 1, Table 2). Of note, these 4 women were amongst the 27 women who received and completed a recall assessment.

### 2.4. Breast Cancer Incidence in NF1 Patients Receiving High-Risk Screening

In the four women diagnosed with breast cancer whilst enrolled in the HR-OBSP at UHN, the age of diagnosis ranged from 44–61 years, and all but one woman reported a family history of breast cancer. All four women harbored different *NF1* variants: one intragenic deletion, splice site, in-frame deletion and frameshift, respectively (Table 3). Through the HR-OBSP, three women had breast abnormalities detected by MRI only (normal mammograms) and one woman had breast abnormalities detected by both mammogram and MRI (Table 3). Since their diagnosis, three women have resumed breast cancer screening with HR-OBSP; the remaining woman underwent a contralateral prophylactic mastectomy and no longer requires high-risk breast screening. Initial abnormal radiological findings for the four women diagnosed with breast cancer were visualized by MRI and appeared as: asymmetric non-mass enhancements or asymmetric small enhancing nodules with irregular shapes and non-circumscribed margins. In all four cases, a second MRI, ultrasound and/or guided core biopsy was recommended to confirm origin of the lesions.

### 2.5. Family History in NF1 Patients Enrolled in High-Risk Screening

Seventeen (28%) of the 61 women enrolled in screening through the HR-OBSP had at least one first and/or second degree relative with breast cancer. Moreover, 37/61 (61%) women were presumed or confirmed de novo cases of NF1: 22/61 (36%) as familial NF1 and 2/61 (3%) as indeterminate. There was no reported genetic and/or clinical diagnosis of NF1 in any of the first or second-degree relatives diagnosed with breast cancer (Table 3).

### 2.6. HR-OBSP Screen Scheduling

Of 60 women who were scheduled at least one breast screen (i.e., imaging modality) with the HR-OBSP, 26/60 (43%) rescheduled at least once. Breast MRIs were the most likely to be rescheduled, with a total of 26 rescheduling events across all 60 women who were scheduled for high-risk breast screening, compared to five events for mammograms and seven for breast ultrasounds. Although 19/60 (32%) women rescheduled their MRIs, only 4/19 (21%) rescheduled ≥2 times. 14/19 (74%) women eventually completed their high-risk MRI screen, leaving only 5/19 (26%) who did not complete their rescheduled MRIs at all.

## 3. Discussion

This 4-year retrospective chart review investigated the uptake of high-risk breast screening, nature of radiological findings and breast cancer incidence in a cohort of female NF1 patients referred to the High-Risk Ontario Breast Screening Program (HR-OBSP) at a tertiary cancer centre in Ontario (UHN). In the province of Ontario, high risk screening is available through the HR-OBSP and entails yearly mammograms and breast MRIs [25] which when combined, have been reported to have the highest predictive value for breast cancer detection [27]. Through the use of this program, we comment on cases of breast cancer in female NF1 patients identified through the HR-OBSP between 2014 and 2018. We examine relevant and noteworthy radiological findings and features of mammary tumors detected during the screening process. According to Frayling et al., the nature of NF1 variant can act as a determinant of breast cancer risk in NF1. Specifically, certain point mutations were shown to significantly increase the risk of breast cancer in female NF1 patients <50 years of age [23]. The mechanism of this effect however, remains unclear. No overlap was found between the type of variants identified in our cohort and those examined by Frayling and colleagues [23]. This can however, be attributed to small cohort sizes in both studies. As more NF1 patients and *NF1* variants are investigated, our understanding of genotype-phenotype correlations in NF1 will improve, and ultimately enhance the clinical management of this patient population. Our study highlights the utility of annual high-risk breast screening in this population of women and underscores a complex need for cross-specialty coordination and benefits of precautionary screening practices in the NF1 patient population. To our knowledge, this the first study of high-risk breast screening uptake in female NF1 patients, and the first to report cases of breast cancer in an Ontario NF1 cohort. Overall, high-risk screen uptake was high, with 95% and 97% of patients completing primary and secondary screens, respectively. Importantly, 100% of women completed recall assessments following abnormal breast imaging and 7% were ultimately diagnosed with breast cancer.

A total of four invasive breast cancers were detected by breast MRI in our cohort, with none being detected by mammography. This finding is in accordance with a recent study in a Canadian high-risk population [26] demonstrating how mammography alone added no value in addition to MRI in the screening detection of breast cancer. In our cohort, three women were diagnosed with breast cancer at ≤45 years of age; all of which were grade II or III. This is in line with previous studies demonstrating a higher prevalence of grade II and III breast cancers diagnosed among women with NF1 between the ages of 40–60 [7,8]. Moreover, given that breast cancer was detected at a later age (>50 years) in one woman via breast MRI, this suggests that high-risk breast screening beyond age 50 may be warranted in this patient population. Interestingly, three of the four women diagnosed with breast cancer in our NF1 cohort had a positive family history of breast cancer. While less than 30% of our cohort reported a family history of breast cancer, 49% of women who completed primary and secondary screening had abnormalities warranting additional workups. This highlights the importance of performing high-risk screening in this population, irrespective of family history. That being said, our study cohort was small, and minimal information exists in the current literature concerning the role of breast cancer family history and breast cancer risk in female NF1. Considering that a positive family history of breast cancer can significantly increase an individual’s lifetime risk [28,29], further investigations are needed in larger NF1 cohorts to examine the moderating effect of family history on NF1-related breast cancer risk.

The majority (74%) of all radiological findings in our NF1 cohort were detected by breast MRI, which is not surprising considering that the detection sensitivity of MRI is >4-fold higher than mammography [30], [31]. Moreover, MRI-based recall rates were higher than mammography-based recall rates within our cohort. Our findings are in line with those of Chiarelli et al., who also examined the effectiveness of high-risk breast screening in women at >25% risk for breast cancer [27]. In this study, Chiarelli and colleagues found recall rates to be highest amongst breast MRIs alone, resulting in 65% of breast cancers within their cohort being identified with MRI alone and the remaining 35% with a combination of MRI and mammograms [27]. Reports from another recent multicenter cohort also demonstrated that MRIs are more sensitive than mammograms for cancer detection in dense breast tissue, which is more prevalent in younger women [32]. This is important for women in our cohort, given their young age (average of 42 years) and the presence of cutaneous neurofibromas. Although the use of breast MRIs has been suggested as a way to subvert the limitations of mammography in NF1 patients, only one case report has been published to date documenting the appearance of neurofibromas on a breast MRI [33]. In our cohort, all cases of breast cancer were detected by breast MRI and the majority of abnormal diagnostic screens were clarified using breast MRI and/or ultrasound, providing evidence for the utility of a combinatorial high-risk screening program for NF1 patients.

Despite the emphasis of previous reports on the importance of breast cancer surveillance and follow-up in women with NF1 [9,12,13,14,34], the efficacy and uptake of such screening programs in this patient population remains unclear. Furthermore, despite studies reporting impaired emotional functioning among NF1 patients, namely higher levels of depression, anxiety and lower levels of self-esteem, [20,22,24] there exists a gap in the literature regarding the participation of NF1 patients in regular clinical surveillance and follow-up. Through referral of female NF1 patients into the HR-OBSP, we observed that 95% of women in our cohort participated in the screening process and completed at least one high-risk screen, with few refusing to complete any screen at all. Interestingly, this level of screening uptake was higher than other reported levels in female carriers of pathogenic variants in *BRCA1/2* [35,36,37,38], another population of women at high-risk of developing breast cancer. This suggests that women diagnosed with NF1 can be integrated into routinized breast screening programs, much like other women at high risk of breast cancer. Moreover, our findings suggest that NF1 patients fully participate in high-risk screening programs that entail regular surveillance and scheduling, despite their potential for increased psychological distress that may negatively impact their ability to participate regularly in clinical appointments and long-term disease management.

A cross-specialty screening methodology that combines breast MRI, mammogram and ultrasound may represent a feasible strategy to screen women at high risk of breast cancer, specifically NF1 patients. Indeed, when combined with clinical breast examinations (CBE), these three screening modalities were found to yield a sensitivity of 95% compared to 45% for mammography and CBE combined in women at >25% risk of breast cancer [30]. For women with NF1, a combinatorial screening strategy may help to overcome the radiological interference of cutaneous neurofibromas that may confound the interpretation of certain screens. Findings from each screen may serve to outweigh the limitations of the other and provide the most complete diagnostic information.

## 4. Materials and Methods

### 4.1. Patient Population

Adults with NF1 seen between January 2014 and December 2018 were identified and taken from the ERNC at UHN using the ERNC clinical database. Eligible participants met the following inclusion criteria: (1) female, (2) age 30–69 years, (3) identified pathogenic variant in NF1. Participants were excluded if they (1) have previously undergone bilateral mastectomies, (2) elected to have HR-OBSP screening outside of UHN, (3) had a diagnosis of breast cancer prior to HR-OBSP referral or (4) tested negative for pathogenic variants in *NF1*. Enrolment of NF1 patients into the HR-OBSP is limited to patients with a confirmed pathogenic variant in *NF1*. Based on the nature of this study, patients with only a clinical diagnosis of NF1 and/or who tested negative for pathogenic *NF1* were excluded. Methods for *NF1* germline genetic analysis included Sanger sequencing, multiplex ligation probe amplification, next generation sequencing, and/or RNA sequencing for deep intronic variants. Genetic testing was administered based on the availability of genetic testing technology at the time of diagnosis.

### 4.2. Data Collection and Analysis

A retrospective chart review was undertaken to identify cases of breast cancer in women diagnosed with NF1 seen at the ERNC between January 2014 and December 2018 and evaluate the uptake and utility of high-risk breast cancer screening in this patient population. The following steps were taken: (1) Identify participants: A search of the ERNC database was conducted to identify women who met eligibility criteria. (2) Determine if screening has been completed: A review of the patient’s electronic medical record was conducted, including imaging reports and consultation notes from the UHN HR-OBSP Nurse Navigator (3) Identify cases of breast cancer and examine the utility of high-risk screening: A review of the patient’s electronic record was conducted, including pathology reports, breast imaging reports. A review of individual patient electronic medical records was conducted to collect information regarding patient date of birth, NF1 variant, date of referral for high risk screening, date of consultation with the Nurse Navigator, date of mammogram, and date of breast MRI. The imaging data was assessed using capture BI-RADS Breast Imaging Reporting and Data System [39] assessment and recommendation categories assigned by the interpreting radiologist for each mammogram, screening ultrasound or breast MRI. For the purposes of this study, we created an initial overall assessment for the screening examination, using the most serious BI-RADS Breast Imaging Reporting and Data System assessment according to the following hierarchy: negative, 1; benign, 2; probably benign, 3; needs additional evaluation, 0; suspicious, 4; and highly suggestive of malignancy, 5. We followed ACR American College of Radiology BI-RADS Breast Imaging Reporting and Data System 5th edition definitions for all metrics. As per BI-RADS Breast Imaging Reporting and Data System audit rules, any mammogram with a BI-RADS Breast Imaging Reporting and Data System 6 assessment (known breast cancer) was excluded from analyses. If applicable, information regarding breast cancer diagnosis, including modality of detection, pathology, and treatment, was also collected through a review of the patient’s electronic record. Descriptive statistics were used to describe clinical findings in the patient population and characteristics of any breast cancer identified. Data collection was undertaken with approval by the UHN Cancer Registry and Data Access committee, Research Ethics Board and Research Quality Integration committee.

## 5. Conclusions

Although our study focused on NF1 patient screening uptake exclusively within the context of a pre-existent government-funded high-risk breast screening program (HR-OBSP), our findings illustrate the potential of a multidisciplinary and regular breast screening program to act as a feasible clinical strategy for cancer surveillance in this underserved patient population. The implications of our findings however, are limited due to our small sample size, exclusion of gene negative NF1 patients, and retrospective design. To shed insight into the long-term sustainability and cost-effectiveness of an integrated screening approach for this unique patient population, we recommend a long-term multi-centre prospective study be conducted to examine high-risk breast screening uptake in a larger cohort of all women diagnosed with NF1.

## Figures and Tables

**Figure 1 cancers-11-00707-f001:**
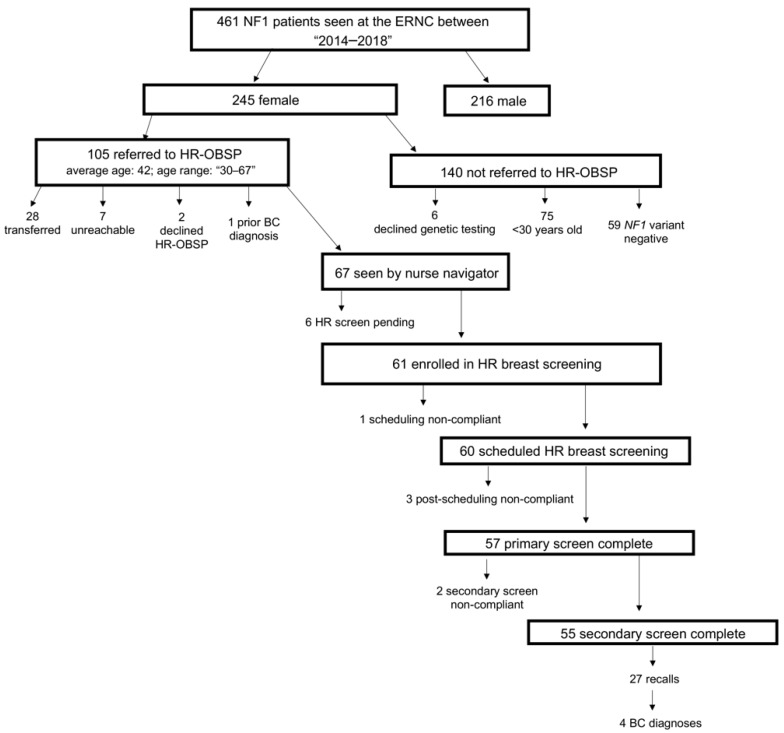
NF1 patient flow chart. NF1: Neurofibromatosis Type I; ERNC: Elizabeth Raab Neurofibromatosis Centre; HR-OBSP: High-Risk Ontario Breast Screening Program; BC: Breast Cancer.

**Table 1 cancers-11-00707-t001:** Summary of the imaging actionable findings distributed by modality in women diagnosed with NF1 seen at the ERNC between 2014–2018.

ACR BI-RADS Descriptor	Screening Mammography (n = 112)	Screening Breast Ultrasound (n = 3)	Screening Breast MRI (n = 108)	X-ray Guided Biopsy (n = 2)	US-Guided Biopsy (n = 10)	MRI-Guided Biopsy (n = 3)
Mass lesion	1 (10%)	-	8 (30%)	-	8 (80%)	1 (33%)
Focus of enhancement	-	-	2 (7%)	-	-	1 (33%)
Non-mass enhancement	-	-	8 (30%)	-	1 (10%)	1 (33%)
Calcification	4 (40%)	-	-	2 (100%)	-	-
Distortion	1 (10%)	-	-	-	1 (10%)	-
Asymmetry	3 (30%)	-	-	-	-	-
Other	1 (10%)		9 (33%)			
**TOTAL**	10 (100%)	-	27 (100%)	2 (100%)	10 (100%)	3 (100%)

^1^ ACR: American College of Radiology, BI-RADS: Breast Imaging Reporting and Data System, US: Ultrasound.

**Table 2 cancers-11-00707-t002:** High-risk OBSP screening uptake by women diagnosed with NF1.

	Total n (%)
**NF1 OBSP Referrals seen by nurse navigator**	67
*Average age at referral*	40
Median	40
Range	30–67
**Scheduled in HR breast screening**	60 (89)
Primary Screen Compliant	57 (95)
Secondary Screen Compliant ^†^	55 (97)
Recalls	27 (49)
Recall Compliant *	27 (100)
Breast Cancer Diagnoses **	4 (7)

^†^ Number of NF1 patients who completed their secondary screen out of the total NF1 patients who completed their primary screens. * Number of NF1 patients who completed their recall assessments out of the total NF1 patients recalled after completing their secondary high-risk breast screens. ** Number of breast cancer diagnoses out of the total NF1 patients who completed at least one screen through the HR-OBSP.

**Table 3 cancers-11-00707-t003:** Identified breast cancer cases in women with confirmed pathogenic variants in *NF1*.

*NF1* Variant	Age at Referral	BC FamilyHx	Abnormal Diagnostic Screen	Breast Pathology	Surgery	Medical Oncology	Radiation
*NF1* exon 12–13 del	44	≥SDR: 1	Baseline 1st HR MRI OBSP screen	IDC GR III, and DCIS GR II, ER^+^/PR^+^/HER2^−^	Left breast Mx. and right breast prophylactic Mx.	Referred	N/A
*NF1* c.4110 + 1G>T	61	FDR: 1; ≥SDR: 2	Baseline 1st HR MRI OBSP screen	Microinvasive LCIS; apocrine DCIS ER^+^/PR^−^	BilLx	Declined TAM	RT
*NF1* c.4973_4978delTCTATA	45	≥SDR: 1	Baseline 1st HR MRI OBSP screen	IDC GR III, ER^−^/PR^−^/HER2^−^	Lx and SNB	Adjuvant CT	Adjuvant RT
*NF1* c.6792insA	45	None	Baseline 1st HR OBSP MRI and Mammogram screens	IDC GR II, ER^+^/PR^+^/HER2^−^	Left breast Lx and SNB	TAM	RT

NF1: Neurofibromatosis type 1, BC: Breast Cancer, DCIS: Ductal carcinoma in situ, IDC: Invasive ductal carcinoma, LCIS: Lobular carcinoma in situ, ER: Estrogen receptor, PR: Progesterone receptor, HER2: Human epidermal growth factor receptor 2, Lx: Lumpectomy, BiLx: Bilateral lumpectomy, UniMx: Unilateral mastectomy, BilMx: Bilateral mastectomy, SNB: Sentinel Node Biopsy, RT: Radiation therapy, CT: Chemotherapy, TAM: Tamoxifen, FDR: First degree relative, SDR: Second degree relative; N/A: Not applicable; Hx: History.

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
