# Peer review of "NF1 Patients Receiving Breast Cancer Screening: Insights from The Ontario High Risk Breast Screening Program"

_cancers, 2019, doi:10.3390/cancers11050707_

Round 1

Reviewer 1 Report

This is a straightforward, sound, descriptive study which should interest those working in similar fields. The numbers are fairly small, but this is unsurprising for a rare condition like NF1. And it is acknowledged by the authors. As a trivial comment, but a related one, I wouldn't bother with decimal points on percentages (e.g. lines 137, 144, 180). But I'd trust the authors to make these minor changes  (if you editors agree) without troubling with re-reviewing!

Author Response

Thank you for the reviews on our manuscript, “NF1 Patients Receiving Breast Cancer Screening: Insights From an Organized Canadian Breast Screening Program for High Risk Women”. We are delighted to have the opportunity to respond to the reviewers’ comments. Please see our revised manuscript with tracked changes (please view with Simple markup viewing for line numbers).

(Reviewer 1): This is a straightforward, sound, descriptive study which should interest those working in similar fields. The numbers are fairly small, but this is unsurprising for a rare condition like NF1. And it is acknowledged by the authors. As a trivial comment, but a related one, I wouldn't bother with decimal points on percentages (e.g. lines 137, 144, 180). But I'd trust the authors to make these minor changes (if you editors agree) without troubling with re-reviewing!

Author response:  Thank you for this comment; we have removed decimal points on percentages throughout the text.

We thank you for your consideration and insightful comments on this manuscript. We hope these responses will address the reviewers concerns and render this version acceptable for publication in Cancers.

Sincerely,

Raymond H. Kim, MD/PhD, FRCPC, FCCMG

Medical Geneticist, University Health Network

Reviewer 2 Report

Although the findings described in the manuscript entitled “NF1 Patients Receiving Breast Cancer Screening: Insights From an Organized Canadian Breast Screening Program for High-Risk Women” are potentially interesting, the manuscript is poorly written and organized.

TITLE

Title should be more specific. Why not mentioning High-Risk Ontario Breast Screening Program

 instead of “an Organized Canadian Breast Screening Program”?

ABSTRACT

Abstract in its present form doesn’t provide an adequate overview of the study.

The authors didn’t state why it was important “to examine the uptake of high-risk breast cancer screening program in NF1 patients”.

Was examining the uptake in breast cancer screening program among NF1 patients the only goal of the study?

Given that the scale of the study was rather modest, it seems premature to state that “… findings support the enrolment of this specific population into breast surveillance programs worldwide”.

RESULTS

Fifty-nine women who were diagnosed with NF1 were negative in the NF1 genetic testing and were excluded from referral to HR-OBSP. NF1 diagnosis is established based on clinical findings and the presence of an NF1 pathogenic variant is not required. Moreover, NF1 genetic testing is difficult and not 100% sensitive. Authors should discuss that exclusion criterion.

Lines 129-136:

Out of 61 women with NF1 enrolled in the HR-OBSP, a total of 95 typically benign or  incomplete screening for diagnostic assessment were reported during the screening period and 37  (38.9%) were considered actionable, required further imaging investigations and/or percutaneous  breast biopsies. Of the total 95 imaging findings, the majority (74%) were described by MRI, 2% by  ultrasound and 7% by mammography. The most common imaging finding reported was  well-circumscribed cutaneous masses in all imaging modalities. The presence of masses in the  fibroglandular breast tissue with either a solid or cystic nature were the second more frequent imaging finding reported.

The authors switch from patient numbers to screening numbers. “ …and 37  (38.9%) were considered actionable…”    37 screenings in how many patients? It’s difficult to understand what the authors are trying to say here.

Lines 137-138:

Of the 37 actionable imaging findings, 15 (40.5%) received a percutaneous imaging-guided breast biopsy.

Again, the wording is very confusing. Who or what received breast biopsy? These paragraphs (lines 128-158) should be carefully re-written in a clear and comprehensible way.

2.4. Breast Cancer Incidences in NF1 Patients Receiving High-Risk Screening section and Table 3

The authors listed constitutive NF1 mutations in four patients who were diagnosed with breast cancer, but never mentioned or discussed these mutations in the text. They cited a recent paper of Frayling and co-authors in the Introduction, but never compared their own findings with the paper’s finding. This would be especially interesting given the data reported by the Frayling et al. group.

Interestingly, Frayling et al., provided evidence that breast cancer risk in NF1 patients may be limited to certain NF1 variants [23], suggesting that a only a subset of women may require enhanced screening.

Also, there’s a typo in this sentence.

Lines 216-223:

2.6. HR-OBSP Screening Uptake  

Of 60 women who were scheduled at least one breast screen (i.e. imaging modality) with the  HR-OBSP, 26/60 (43.3%) rescheduled at least once. Breast MRIs were the most likely to be rescheduled, with a total of 26 rescheduling events across all 60 women who were scheduled for  high-risk breast screening, compared to 5 events for mammograms and 7 for breast ultrasounds. Although 19/60 (31.6%) women rescheduled their MRIs, only 4/19 (21.0%) rescheduled  _2 times.  14/19 (73.6%) women eventually completed their high-risk MRI screen, leaving only 5/19 (26.3%)  who did not complete their rescheduled MRIs at all.

There is ambiguity in how the authors use terms “compliance” and “uptake” throughout the text. These two terms are used either interchangeably, or as distinct entities. Based on the paragraph shown above, one might think that screening uptake is about rescheduling screening events. Authors need to clarify their use of “compliance” and “uptake”.

DISCUSSION

Lines 237-238:

Overall, compliance was high, with 95.0% and 96.5% of patients completing primary and secondary screens, respectively.

Lines 281-283:

Through referral of female NF1 patients into the HR-OBSP, we observed that 91.9% of women in our cohort were compliant throughout the screening process and completed at least one high-risk screen, with few refusing to complete any screen at all.

It’s unclear where 91.9% comes from.

Author Response

Thank you for the reviews on our manuscript, “NF1 Patients Receiving Breast Cancer Screening: Insights From an Organized Canadian Breast Screening Program for High Risk Women”. We are delighted to have the opportunity to respond to the reviewers’ comments. Please see our revised manuscript with tracked changes (please view with Simple markup viewing for line numbers).

(Reviewer 2): Although the findings described in the manuscript entitled “NF1 Patients Receiving Breast Cancer Screening: Insights From an Organized Canadian Breast Screening Program for High-Risk Women” are potentially interesting, the manuscript is poorly written and organized.

TITLE

(Reviewer 2): Title should be more specific. Why not mentioning High-Risk Ontario Breast Screening Program instead of “an Organized Canadian Breast Screening Program”?

Author response:  We thank you for this comment and have changed the title of our manuscript to reflect the specific nature of our study.

ABSTRACT

(Reviewer 2): Abstract in its present form doesn’t provide an adequate overview of the study. The authors didn’t state why it was important “to examine the uptake of high-risk breast cancer screening program in NF1 patients”.

Author response: We thank you for this comment and agree that our abstract did not provide an adequate overview of our study. We have made sure to clarify our study methodology and reasoning in the abstract in order to better reflect the purpose of our study.

(Revision Lines 28-36) Women with NF1 <50 years of age possess an up to five-fold increased risk of developing breast cancer compared with the general population. Impaired emotional functioning is reported as a comorbidity that may influence the participation of NF1 patients in regular clinical surveillance despite their increased risk of breast and other cancers. Despite emphasis on breast cancer surveillance in women with NF1, the uptake and feasibility of high-risk screening programs in this population remains unclear. A retrospective chart review between 2014-2018 of female NF1 patients seen at the Elizabeth Raab Neurofibromatosis Clinic (ERNC) in Ontario was conducted to examine the uptake of high risk breast cancer screening, radiologic findings, and breast cancer characteristics.

(Reviewer 2): Was examining the uptake in breast cancer screening program among NF1 patients the only goal of the study?

Author response: We thank you for this question and agree that the purpose of our study was not clearly stated in the abstract. We have made sure to clarify the goals and purpose of our study in the abstract (Revision lines 28-36). For completeness, we have also clarified the goals of our study in our discussion (Revision lines 247-250).  

(Revision Lines 28-36) Women with NF1 <50 years of age possess an up to five-fold increased risk of developing breast cancer compared with the general population. Impaired emotional functioning is reported as a comorbidity that may influence the participation of NF1 patients in regular clinical surveillance despite their increased risk of breast and other cancers. Despite emphasis on breast cancer surveillance in women with NF1, the uptake and feasibility of high-risk screening programs in this population remains unclear. A retrospective chart review between 2014-2018 of female NF1 patients seen at the Elizabeth Raab Neurofibromatosis Clinic (ERNC) in Ontario was conducted to examine the uptake of high risk breast cancer screening, radiologic findings, and breast cancer characteristics.

(Revision lines 247-250: This 4-year retrospective chart review investigated the uptake of high-risk breast screening, nature of radiological findings and breast cancer incidence in a cohort of female NF1 patients referred to the High-Risk Ontario Breast Screening Program (HR-OBSP) at a tertiary cancer centre in Ontario (UHN).

(Reviewer 2): Given that the scale of the study was rather modest, it seems premature to state that “… findings support the enrolment of this specific population into breast surveillance programs worldwide”.

Author response: Thank you for this comment, and we entirely agree with this comment from the reviewer. We have made changes to this final sentence of our abstract (revision lines 38-39) to better reflect the applicability of our findings. 

(Revision lines 38-39): Our findings support the integration of a formal breast screening programs in clinical management of NF1 patients.

RESULTS

(Reviewer 2): Fifty-nine women who were diagnosed with NF1 were negative in the NF1 genetic testing and were excluded from referral to HR-OBSP. NF1 diagnosis is established based on clinical findings and the presence of an NF1 pathogenic variant is not required. Moreover, NF1 genetic testing is difficult and not 100% sensitive. Authors should discuss that exclusion criterion.

Author response: Thank you for your comment. We have included an explanation of our exclusion criteria in our introduction (revision lines 106-108), our results section (see: 2.1. Patient Population revision lines 114-121) and our methods section (see: 4.1 Patient population revision lines 339-346) to clarify and better explain the nature of our exclusion criteria.

(Revision lines 106-108): Enrolment into the HR-OBSP is limited to confirmed carriers of pathogenic variants. Consequently, patients with solely a clinical diagnosis and/or no pathogenic NF1 gene variant are not screened through the HR-OBSP.

(Revision lines 114-121): Of the 140 (57%) women not referred to HR-OBSP, 75 were <30 years of age. As enrolment into the HR-OBSP is limited to patients with a confirmed pathogenic variant in NF1, 6 women were excluded from the study because they declined genetic testing and 59 women were excluded because they tested negative for pathogenic variants in NF1. Methods for NF1 germline genetic analysis included Sanger sequencing, multiplex ligation probe amplification, next generation sequencing, and/or RNA sequencing for deep intronic variants. Genetic testing was administered based on the availability of genetic testing technology at the time of diagnosis.

(Revision lines 340-347): or 4) tested negative for pathogenic variants in NF1. Enrolment of NF1 patients into the HR-OBSP is limited to patients with a confirmed pathogenic variant in NF1. Based on the nature of this study, patients with only a clinical diagnosis of NF1 and/or who tested negative for pathogenic NF1 were excluded.  Methods for NF1 germline genetic analysis included Sanger sequencing, multiplex ligation probe amplification, next generation sequencing, and/or RNA sequencing for deep intronic variants. Genetic testing was administered based on the availability of genetic testing technology at the time of diagnosis.

(Reviewer 2): Lines 129-136:

Out of 61 women with NF1 enrolled in the HR-OBSP, a total of 95 typically benign or incomplete screening for diagnostic assessment were reported during the screening period and 37 (38.9%) were considered actionable, required further imaging investigations and/or percutaneous breast biopsies. Of the total 95 imaging findings, the majority (74%) were described by MRI, 2% by ultrasound and 7% by mammography. The most common imaging finding reported was well-circumscribed cutaneous masses in all imaging modalities. The presence of masses in the fibroglandular breast tissue with either a solid or cystic nature were the second more frequent imaging finding reported.

The authors switch from patient numbers to screening numbers. “… and 37 (38.9%) were considered actionable…”    37 screenings in how many patients? It’s difficult to understand what the authors are trying to say here.

Author response: Thank you for this comment, and we agree that the wording of these paragraphs needs to be clarified. We have made changes to the wording and organization of this paragraph to reflect a clearer explanation of the imaging findings (Revision lines 139-143; Revision lines 143-144; Revision lines 149-150).

(Revision lines 139-143): A total of 95 imaging findings classified as “typically benign” or “incomplete screening for diagnostic assessment”, were reported during the screening period amongst our cohort of 61 NF1 patients enrolled in the HR-OBSP. 37/95 (39%) of these imaging findings were considered to be actionable. Actionable is defined as an imaging finding that requires further imaging investigations and/or percutaneous breast biopsies.

(Revision lines 143-144): Of the total 95 imaging findings, the majority 70/95 (74%) were described by MRI, 2/95 (2%) by ultrasound and 7/95 (7%) by mammography.

(Revision lines 149-150): Of the 37 actionable imaging findings, 15/37 (41%) findings were further evaluated by percutaneous imaging-guided breast biopsy.

(Reviewer 2): Lines 137-138:

Of the 37 actionable imaging findings, 15 (40.5%) received a percutaneous imaging-guided breast biopsy.

Again, the wording is very confusing. Who or what received breast biopsy? These paragraphs (lines 128-158) should be carefully re-written in a clear and comprehensible way.

Author response: Thank you for this suggestion. We have made sure to carefully re-word and reorganize this paragraph to reflect a clearer explanation of the imaging findings (Revision lines 149-150).

(Revision lines 149-150): Of the 37 actionable imaging findings, 15/37 (41%) findings were further evaluated by percutaneous imaging-guided breast biopsy.

(Reviewer 2): 2.4. Breast Cancer Incidences in NF1 Patients Receiving High-Risk Screening section and Table 3

The authors listed constitutive NF1 mutations in four patients who were diagnosed with breast cancer, but never mentioned or discussed these mutations in the text. They cited a recent paper of Frayling and co-authors in the Introduction, but never compared their own findings with the paper’s finding. This would be especially interesting given the data reported by the Frayling et al. group.

Interestingly, Frayling et al., provided evidence that breast cancer risk in NF1 patients may be limited to certain NF1 variants [23], suggesting that a only a subset of women may require enhanced screening.

Also, there’s a typo in this sentence.

Author response: Thank you for this suggestion and we agree that comparing our findings with that of Frayling et al would be of particular interest. We have made sure to compare our genetic findings with that of those reported in the Frayling et al paper and have incorporated that information into the discussion section of our manuscript (Revision lines 255-262). We have also commented on the nature of the variants in revision lines 213-214. We have addressed the typo in the above-mentioned sentence (Revision lines 83-85).

(Revision lines 255-262): According to Frayling et al., the nature of NF1 variant can act as a determinant of breast cancer risk in NF1. Specifically, certain point mutations were shown to significantly increase the risk of breast cancer in female NF1 patients <50 years of age [23]. The mechanism of this effect however, remains unclear. No overlap was found between the type of variants identified in our cohort and those examined by Frayling and colleagues [23]. This can however, be attributed to small cohort sizes in both studies. As more NF1 patients and NF1 variants are investigated, our understanding of genotype-phenotype correlations in NF1 will improve, and ultimately enhance the clinical management of this patient population.

(Revision lines 213-214): All four women harbored different NF1 variants: one intragenic deletion, splice site, in-frame deletion and frameshift, respectively. (Table 3).

(Revision lines 83-85): Interestingly, Frayling et al., provided evidence that breast cancer risk in NF1 patients may be limited to certain NF1 variants [23], suggesting that only a subset of women may require enhanced screening.

(Reviewer 2): Lines 216-223:

 2.6. HR-OBSP Screening Uptake  

Of 60 women who were scheduled at least one breast screen (i.e. imaging modality) with the HR-OBSP, 26/60 (43.3%) rescheduled at least once. Breast MRIs were the most likely to be rescheduled, with a total of 26 rescheduling events across all 60 women who were scheduled for high-risk breast screening, compared to 5 events for mammograms and 7 for breast ultrasounds. Although 19/60 (31.6%) women rescheduled their MRIs, only 4/19 (21.0%) rescheduled ³ 2 times.  14/19 (73.6%) women eventually completed their high-risk MRI screen, leaving only 5/19 (26.3%) who did not complete their rescheduled MRIs at all.

There is ambiguity in how the authors use terms “compliance” and “uptake” throughout the text. These two terms are used either interchangeably, or as distinct entities. Based on the paragraph shown above, one might think that screening uptake is about rescheduling screening events. Authors need to clarify their use of “compliance” and “uptake”.

Author response: We thank you for this suggestion and agree that we were ambiguous in our use of the words “compliance” and “uptake”. We have clarified our ambiguous use of “compliance” and “uptake” by choosing to refer to uptake throughout the text as a consistent reference to the completion of screening. We have equally modified our discussion of screen scheduling to discuss solely that of rescheduling screening events in an effort to further clarify our findings. (See revision lines: 40, 92, 108, 175, 176, 183, 195, 238, 247, 265, 267, 314, 316, 320, and 376)

DISCUSSION

(Reviewer 2): Lines 237-238: 

Overall, compliance was high, with 95.0% and 96.5% of patients completing primary and secondary screens, respectively.

Lines 281-283:

Through referral of female NF1 patients into the HR-OBSP, we observed that 91.9% of women in our cohort were compliant throughout the screening process and completed at least one high-risk screen, with few refusing to complete any screen at all. 

It’s unclear where 91.9% comes from.

Author response: We thank you for this comment and agree that 91.9% is misplaced in this paragraph. We have made sure to change 91.9% to 95% to reflect the correct percentage of women with NF1 who completed a primary screen through the HR-OBSP. (Revision line 313-315).

(Revision lines 313-315): Through referral of female NF1 patients into the HR-OBSP, we observed that 95% of women in our cohort participated in the screening process and completed at least one high-risk screen, with few refusing to complete any screen at all.

We thank you for your consideration and insightful comments on this manuscript. We hope these responses will address the reviewers concerns and render this version acceptable for publication in Cancers.

Sincerely,

Raymond H. Kim, MD/PhD, FRCPC, FCCMG

Medical Geneticist, University Health Network

Reviewer 3 Report

The authors present an interesting study on the uptake of the breast cancer screening by women affected by neurofibromatosis type I. The study is retrospective, based on a relatively low number of cases analysed (67 women), but reaches solid conclusions about the high adherence of NF1 women to the screening, confirms the MRI as the best method of analysis in such cases and points to the direction of a truly increased risk of breast cancer, with four cases found out of 67, with an average age of 42 years, confirming previous results.

Nevertheless, despite the small number of patients under study and despite the retrospective design, the paper is very nice to read and gives important information. However, the authors could add a point in the discussion concerning these two main shortcomings of the study, as a caveat for the soundness of their results

Among the selection criteria there is a positive genetic test result, but it is not easy to understand how many were excluded only for this criteria and what method of genetic analysis was used

In the table 3 the relatives with breast cancer had neurofibromatosis or not? It could be important to show in the table and eventually comment

Author Response

Thank you for the reviews on our manuscript, “NF1 Patients Receiving Breast Cancer Screening: Insights From an Organized Canadian Breast Screening Program for High Risk Women”. We are delighted to have the opportunity to respond to the reviewers’ comments. Please see our revised manuscript with tracked changes (please view with Simple markup viewing for line numbers).

(Reviewer 3): The authors present an interesting study on the uptake of the breast cancer screening by women affected by neurofibromatosis type I. The study is retrospective, based on a relatively low number of cases analysed (67 women), but reaches solid conclusions about the high adherence of NF1 women to the screening, confirms the MRI as the best method of analysis in such cases and points to the direction of a truly increased risk of breast cancer, with four cases found out of 67, with an average age of 42 years, confirming previous results.

Author response: No response required.

(Reviewer 3): Nevertheless, despite the small number of patients under study and despite the retrospective design, the paper is very nice to read and gives important information. However, the authors could add a point in the discussion concerning these two main shortcomings of the study, as a caveat for the soundness of their results

Author response: We thank you for this suggestion and agree that these two limitations of our study should be addressed. We have made an addition to the body of our conclusions paragraph (see: 5. Conclusions, Revision lines 379-385) to discuss the limitations of our study and our suggestions for improving the soundness of our findings in future studies.

(Revision lines 379-385): The implications of our findings however, are limited due to our small sample size, exclusion of gene negative NF1 patients, and retrospective design.  To shed insight into the long-term sustainability and cost-effectiveness of an integrated screening approach for this unique patient population, we recommend a long-term multi-centre prospective study be conducted to examine high-risk breast screening uptake in a larger cohort of all women diagnosed with NF1.

(Reviewer 3): Among the selection criteria there is a positive genetic test result, but it is not easy to understand how many were excluded only for this criteria and what method of genetic analysis was used

Author response: Thank you for this comment. We have made an addition to the text of 2.1. Patient Population, to further clarify our exclusion criteria, and the number of women who were excluded based on this criterion (Lines 114-121). We have also added the method of genetic analysis that was used. For completeness, we have also included information on our exclusion criteria in our methods section (see: 4.1. Patient Population, lines 340-347).  We have also further commented on criteria for referral to the HR-OBSP in revision lines 106-108 of our introduction.

(Revision lines 114-121): Of the 140 (57%) women not referred to HR-OBSP, 75 were <30 years of age. As enrolment into the HR-OBSP is limited to patients with a confirmed pathogenic variant in NF1, 6 women were excluded from the study because they declined genetic testing and 59 women were excluded because they tested negative for pathogenic variants in NF1. Methods for NF1 germline genetic analysis included Sanger sequencing, multiplex ligation probe amplification, next generation sequencing, and/or RNA sequencing for deep intronic variants. Genetic testing was administered based on the availability of genetic testing technology at the time of diagnosis.

(Revision lines 340-347): or 4) tested negative for pathogenic variants in NF1. Enrolment of NF1 patients into the HR-OBSP is limited to patients with a confirmed pathogenic variant in NF1. Based on the nature of this study, patients with only a clinical diagnosis of NF1 and/or who tested negative for pathogenic NF1 were excluded.  Methods for NF1 germline genetic analysis included Sanger sequencing, multiplex ligation probe amplification, next generation sequencing, and/or RNA sequencing for deep intronic variants. Genetic testing was administered based on the availability of genetic testing technology at the time of diagnosis.

(Revision lines 106-108): Enrolment into the HR-OBSP is limited to confirmed carriers of pathogenic variants. Consequently, patients with solely a clinical diagnosis and/or no pathogenic NF1 gene variant are not screened through the HR-OBSP

(Reviewer 3): In the table 3 the relatives with breast cancer had neurofibromatosis or not? It could be important to show in the table and eventually comment

Author response: We thank you for this suggestion and have made sure to include this information in the body of paragraph 2.5. Family History in NF1 Patients Enrolled in High-Risk Screening (Revision lines 235-237):

(Revision lines 235-237): There was no reported genetic and/or clinical diagnosis of NF1 in any of the first or second-degree relatives diagnosed with breast cancer (Table 3)

We thank you for your consideration and insightful comments on this manuscript. We hope these responses will address the reviewers concerns and render this version acceptable for publication in Cancers.

Sincerely,

Raymond H. Kim, MD/PhD, FRCPC, FCCMG

Medical Geneticist, University Health Network

Round 2

Reviewer 2 Report

Looks good now.